# Insights into the Functional Responses of Four Neotropical-Native Parasitoids to Enhance Their Role as Biocontrol Agents Against *Anastrepha fraterculus* Pest Populations

**DOI:** 10.3390/insects16090919

**Published:** 2025-09-02

**Authors:** Segundo Ricardo Núñez-Campero, Lorena del Carmen Suárez, Flávio Roberto Mello Garcia, Jorge Cancino, Pablo Montoya, Sergio Marcelo Ovruski

**Affiliations:** 1La Rioja Regional Center for Scientific Research and Technology Transfer (CRILAR-CONICET), Entre Ríos y Mendoza s/n, Anillaco 5301, La Rioja, Argentina; segundo.nc@conicet.gov.ar; 2Department of Exact, Physical and Natural Sciences, Institute of Conservation Biology and Paleobiology (IBICOPA), National University of La Rioja (UNLaR), Avenida Luis de la Fuente s/n, Ciudad de La Rioja 5300, La Rioja, Argentina; 3Plant, Animal, and Food Health Bureau of the Government of the San Juan Province, Nazario Benavides 8000 Oeste, Rivadavia 5413, San Juan, Argentina; lorenasuarez@conicet.gov.ar; 4National Scientific and Technical Research Council (CCT CONICET) San Juan, Avenida Libertador General San Martín 1109, San Juan 5400, San Juan, Argentina; 5Department of Ecology, Zoology and Genetics, Institute of Biology, Federal University of Pelotas, Pelotas 96000, RN, Brazil; flavio.garcia@ufpel.edu.br; 6Agricultural Sciences College, Autonomous University of Chiapas, Entronque Carretera Costera y Pueblo de Huehuetán, Huehuetán 30660, Chiapas, Mexico; jorge.cancino@unach.mx; 7Institute of Biosciences, Autonomous University of Chiapas, Boulevard Príncipe Akishino S/N., Col. Solidaridad 2000, Tapachula 30798, Chiapas, Mexico; 8Microbiological Industrial Processes and Biotechnology Pilot Plant (PROIMI-CONICET), Biological Control Department, Avenida Belgrano y Pasaje Caseros, San Miguel de Tucumán 4001, Tucumán, Argentina; sovruski@conicet.gov.ar

**Keywords:** fruit fly parasitoids, South American fruit fly, parasitoid–host interaction, parasitoid selection, host density, fruit fly biological control

## Abstract

*Anastrepha fraterculus* is one of the most damaging Neotropical insect pests to fruit production and trade, including causing quarantine issues for world regions unaffected by this pest. Using indigenous parasitoid species as a potential control tool in *A. fraterculus* integrated management programs (IMPs), where augmentative biological control (ABC) is an eco-friendly strategy that requires selecting potential control agent candidates, functional response analysis can assess the change in the attack rate of a given host by a parasitoid based on its searching, manipulating, and consuming time, providing valuable insights into the most suitable biocontrol agent. We used the “frair” package of R software for functional response estimation, which allowed comparison among the determined models of four Neotropical parasitoid species associated with *A. fraterculus*: the pupal parasitoid *Coptera haywardi* and larval parasitoids *Ganaspis pelleranoi*, *Doryctobracon crawfordi*, and *Opius bellus*. Results showed that *G. pelleranoi* and *C. haywardi* were highly efficient at host utilization in terms of attack rate and discarded hosts, but *G. pelleranoi* showing a shorter host manipulation time and a Type III functional response, thus contributing to better host/parasitoid system stability. Therefore, both parasitoids are potential biocontrol agents in programs involving the ABC strategy.

## 1. Introduction

Tephritidae, known as “true fruit flies,” is a Diptera family with a wide diversity of food habits strongly associated with plant organs, and so some species are pests for the world’s fruit and vegetable industry [1]. In Argentina, the Neotropical-native fruit fly *Anastrepha fraterculus* Wiedemann (Diptera: Tephritidae), or South American Fruit Fly, shares with the African-native *Ceratitis capitata* Wiedemann (Diptera: Tephritidae), or Mediterranean fruit fly, both the central and northern fruit-growing Argentinian regions, where they are major pests of fruit crops [2]. The techniques used to control these two pests of economic and quarantine importance in Argentina were restricted to bait-spray applications and cultural practices, and only in the case of *C. capitata* is the Sterile Insect Technique (SIT) used in some provinces as an environmentally friendly alternative [2,3,4]. However, the use of the SIT against *A. fraterculus* is not yet ready for large-scale use, reducing the alternatives to mass trapping, chemical and cultural controls, and phytosanitary barriers as quarantine protection [5,6]. Moreover, biological control has not yet been used against this fruit fly pest in Argentina [4]. From 1930 onwards, the introduction and release of numerous exotic parasitoid species into Latin America for biological control of various pest species from the Neotropical genus *Anastrepha* Schiner prevailed among all fruit fly integrated management programs [7]. Significant performance against *Anastrepha* species was mainly achieved with the Asian-native braconid larval–pupal parasitoid *Diachasmimorpha longicaudata* (Ashmead), which was successfully established in release areas of many Latin American countries [8,9,10]. Using Neotropical hymenopteran parasitoids instead of exotic species is a valid control technique as an alternative to classical biological control [11,12,13,14,15]. Despite their long-term co-evolutionary history in sympatry with *Anastrepha* pests, native parasitoids received little attention on the assumption that they could not exert economically significant levels of control [13]. Nevertheless, the implementation of augmentative biological control (=ABC) against pest fruit fly species in the Americas [7,9,16] encouraged colonization of Neotropical-native parasitoids naturally associated with the *Anastrepha* genus [13,17,18]. This biological control approach strategically increased the natural mortality of pest species of the *Anastrepha* genus inflicted by indigenous parasitoids through mass release into a target area [19].

As a follow-up to the Mexico initiative, an ongoing field of research in Argentina involves native Neotropical parasitoids for the biological control of the two economically important fruit fly species. Fundamental research on this issue began 15 years ago, involving the figitid species *Ganaspis pelleranoi* (Brèthes), a diapriid parasitoid *Coptera haywardi* (Oglobin), and the larval braconid parasitoids *Doryctobracon crawfordi* (Viereck) and *Opius bellus* Gahan (Braconidae) [4]. Findings demonstrated that some indigenous parasitoid species could be combined with the exotic *D. longicaudata* in augmentative releases in Argentina [20]. In this sense, the next step to be covered is transitioning from small-scale experimental rearing of those native parasitoid species to mass-production scale for open-field augmentative release, providing an eco-friendly alternative for *A. fraterculus* control [4].

Among the different selective control methods to suppress fruit fly pest populations, the mass release of specific and efficient parasitoids for ABC is a highly selective strategy [21,22,23], overcoming the limitations of classical biological control when this method is used against pest fruit flies [24]. Therefore, parasitoid mass release is essential to any area-wide integrated fruit fly management program [19,25]. However, implementing ABC requires understanding the potential performance of parasitoid species selected as augmentative biocontrol agents, which necessarily includes the mass production and large-scale management of both the host species and the parasitoid. Consequently, thorough knowledge of field and quality control parameters is needed to achieve a suitable standard for successfully releasing mass-reared parasitoids [24,26]. Thus, both host and parasitoid biological aspects, such as population parameters, are crucial to assessing the parasitoid’s production cost [27]. In addition, it is necessary to consider that host–parasitoid interaction directly influences demographic parameters, as foraging and reproduction are closely linked to this relationship [28].

The functional response analyzes relevant parameters involved in the complexity of a host–parasitoid trophic interaction. That is because the functional response describes the change in the per capita rate of the consumer (e.g., predator, parasitoid) as a function of the availability of the resource (e.g., prey, host), considering the consumer’s time spent searching for and handling the resource [29,30]. Thus, the functional response can be regarded as the cornerstone for any study of host–parasitoid or prey–predator trophic systems in which the number of hosts parasitized, or prey consumed, will influence the development, survival, and reproduction of the natural enemy population [31,32]. In this regard, functional response measurements help to understand parasitoid/predator–prey relationships, which can be used to design effective and sustainable biocontrol programs supporting the selection of the most suitable natural enemies [33]. Researchers agree that functional response is critical information in many ecological studies, from theoretical [34] to practical application [35,36]. However, little attention has been focused on improving the interpretation of the empirical results of the functional response and on applying and extrapolating such results to field-realistic conditions [37]. Recently, Pritchard et al. [38] developed an R package called “frair” v0.5.100 that allows selection, fitting, and comparison between different functional response models and associated parameters. This package provides a systematic approach to estimating functional response parameters.

Although studies on experimental rearing and demographic parameters of *G. pelleranoi*, *C. haywardi*, *D. crawfordi*, and *O. bellus* have been reported [27,39,40,41], no previous research has assessed the functional response of the four parasitoid species listed above on *A. fraterculus*. Two Neotropical fruit fly parasitoid species, such as *Utetes anastrephae* (Viereck) and *C. haywardi*, have been studied for their functional response and mutual intraspecific interference under mass rearing situations, while using *Anastrepha ludens* (Loew) and *Anastrepha obliqua* McQuart, respectively, as host species [42,43].

Given all the above-described issues, the current study aims to analyze functional response parameters using the “frair” package to provide tools for visualizing their performance as biocontrol agents, and to obtain some insights for the mass production of indigenous parasitoids *C. haywardi*, *G. pelleranoi*, *D. crawfordi*, and *O. bellus*, reared on *A. fraterculus* pupae and larvae. We hypothesize that such functional response analyses can be useful for estimating the parasitoid species potentially most suitable for use as a biological control agent against *A. fraterculus*.

This study’s results stem from ongoing research initiated in 2010, focused primarily on colonization, mass rearing, and augmentative releases of Neotropical-native parasitoid species of fruit fly pest species in Argentina. Therefore, our findings are discussed in the context of selecting effective native species to support environmentally sustainable strategies for controlling *A. fraterculus* populations, with potential implications for broader application in the Americas.

## 2. Materials and Methods

### 2.1. Insect Rearing and Sources

The parasitoids belonging to four hymenopteran species and the tephritid fly *A. fraterculus* were reared at the Biological Control Department from the Biotechnology and Microbiological Industrial Processes Pilot Plant (PROIMI, Spanish acronyms), San Miguel de Tucumán, Argentina. The three larval parasitoid species (*G. pelleranoi*, *D. crawfordi*, and *O. bellus*) and the pupal parasitoid (*C. haywardi*) were successfully reared under artificial conditions (25 ± 1 °C, 70 ± 5% RH, 12:12 h photoperiod) using lab-reared, early-third-instar larvae, and 2 d old pupae of *A. fraterculus*, respectively. Adult parasitoids were provided with honey and water ad libitum and held in 30 × 30 × 30 cm cubical Plexiglass cages covered by white cotton voile cloth on two opposite sides. The hosts were offered to 5–10-day-old parasitoid females for 5–6 h every other day. After exposure, parasitized host larvae and pupae were placed in 500 mL hinged plastic cups with 100 cm^3^ previously sterilized vermiculite (Intersum^®^, Aislater S.R.L., Córdoba, Argentina) on the bottom as a pupation substrate and kept there until the adults emerged. Fly adults were reared in 90 × 15 × 110 cm (length × width × height) rectangular aluminum-framed, voile-cloth-covered cages under the previously described lab conditions. Flies were fed ad libitum with a mixture (1:3, w:w) of yeast hydrolysate enzymatic (MP Biomedicals, LLC, Solon, OH, USA) and refined cane sugar (Ledesma S.A.A.I, Libertador General San Martin, Jujuy, Argentina) placed in 12 × 1 cm (diameter × height) plastic vials; water was provided daily in 200 mL plastic containers with a white cotton wick. When the flies were 10–20 days old, oviposited eggs were collected daily and incubated at 26 °C for four days. The eggs were sown in an artificial diet for larval development based on wheat germ, sugar, brewer’s yeast, citric acid, agar-agar, water, vitamins, minerals, and preserving agents. Larvae were reared in a 28.5 × 19.5 × 2 cm (length × width × height) high-density styrofoam tray. The *C. haywardi*, *A. pelleranoi*, *O. bellus*, and *D. crawfordi* cohorts used in the experiment were at ~48 generations under artificial rearing. Parasitoids were initially collected in Horco Molle, Yerba Buena, Tucumán, Argentina, an area representative of the Yungas biome (Tucuman-Bolivian rainforest), and duly identified by the senior author, S.M.O. Voucher material was placed in the entomological collection of the Miguel Lillo Foundation (FML) in San Miguel de Tucuman, Argentina.

### 2.2. Experimental Setup

Experiments were performed under controlled conditions [25 ± 1 °C, 75% RH, 12:12 h (light/darkness)] inside an airtight chamber for insect rearing (Percival, Model DR36VL, Percival Scientific Inc., Perry, IA, USA). For each treatment, one naïve mated 6 d old female of each parasitoid species was individually placed into an 8 × 10 × 15 cm Plexiglass cage with a cotton voile sleeve on the front face and provided with water and honey as food. The functional response study tested nine different host densities (treatments) supplied to the parasitoids: 1, 5, 10, 15, 20, 40, 60, 80, and 100 larvae or puparia. Naked (no larval diet) host larvae were exposed to a parasitoid female in a 5 cm diameter × 1 cm height double voile mesh-covered dish, where larvae were placed between screens, avoiding larval refuge and allowing contact with the parasitoids. In the case of *C. haywardi*, the *A. fraterculus* puparia were exposed on the bottom of the plastic dish. Host exposure time to parasitoids was 24 h. After exposure, each host larva and puparia cohort was kept in 500 mL hinged plastic cups with 100 cm^3^ vermiculite Intersum^®^ (Aislater S.R.L., Córdoba, Argentina). Each cup was labeled with parasitoid species, host densities, and exposure date, and the cup was kept until the adult insect emerged inside the airtight chamber. The four experiments regarding host depletion assumed that larvae or pupae were parasitized only once, as they were not replaced during the experiment. Corresponding functions for the condition above were selected for the analysis (see Section 2.3). The number and sex of parasitoid offspring and the number of non-emerged puparia were recorded. Each host density was replicated 41, 32, 25, and 21 times for *D. crawfordi*, *G. pelleranoi*, *O. bellus*, and *C. haywardi*, respectively, according to host and parasitoid availability, but ensuring at least 20 replicates per species.

### 2.3. Functional Response Analysis

The functional response of each parasitoid species was estimated using the software R [44] and the specific package “frair” (v0.5.100) that allows selection, fitting, and comparison among standard functional response models and the associated parameters described below [38]. The first step was the model selection, fitting the data sets resulting from assays at different host densities of each species using the function “*frair fit.*” This function fits typical non-linear consumer–resource (host–parasitoids) curves to integer data using maximum likelihood estimation, implementing the “*mle2*” function from the “*bbmle*” package [45]. This function applies the method “Nelder-Mead,” with 5000 maximum iterations. The model selections were performed considering that experiments were carried out with resource depletion, which means that the host parasitized during the experiment was not replaced. A generalized form of the traditional functional response family models, the Holling disc equation (Equation (1)) [46], was fitted, providing a solution by integrating the instantaneous attack rate over time (Equation (2)) [47,48].(1)$$N_{e}=\frac{a\;T\;N^{\left(q+1\right)}}{1+a\;h\;N^{\left(q+1\right)}}$$(2)$$N_{e}=N_{0}(1-\exp\left(a\;N_{0}^{q}\;\left(h\;N_{e}-T\right)\right)$$
where *N_e_* is the expected number of the attacked hosts, *N*_0_ (equal to *N* in Equation (1)) is the number of initial experimental host densities, *T* is the experimental time (fixed value), *a* is the capture rate (here follows the relationship shown in Equation (3), with *X* = *N*_0_ and *b* constant), *h* is the handling time, and *q* is a scaling exponent defining the functional response type (Type II: *q* = 0, Type III: *q* > 0, and Type I: *q* and *h* = 0).(3)$$a=bX^{q}$$

The second step was the model fitting using the “*frair_fit*” function that provides the algorithms to solve Equation (2), using the “Lambert-W” function (W) because the expected number of hosts attacked (*N_e_*) is on both sides of the equation. The data were modeled with the function “*flexpnr*” (for no replacement data set) that provides a solution to the problem of depletion by integrating instantaneous consumption over time from the package “*frair*”. The “*flexpnr*” is a flexible function that allows responses of Type II and III, depending on the significance of the scaling exponent (*q*) value. The responses with *q* values with significant differences from 0 were fitted to Hassell’s Type III functional response model without assuming replacement (“*HassIIInr*”). In this function, the quantity of prey consumed (*Ne*) adheres to the same pattern established for Roger’s Type II response. However, the capture rate (*a*) fluctuates with prey density, following a hyperbolic relationship (Equation (4)).(4)$$a=\frac{b\;X}{1+c\;X}\;$$

The model comparison was the third step, in which the models generated by each parasitoid species under study were compared and selected using the Akaike Information Criterion (AIC). No direct statistical comparison between different parasitoid functional responses was possible, so *O. bellus* was not included in the statistical comparative analysis [49]. Instead, Pritchard et al. [38] recommend using the confidence intervals (CIs) of the parameters of a model as an equivalent method for a null hypothesis test. CIs measure the uncertainty of the parameter estimates and are calculated using the model’s degree of freedom (df) [38]. Therefore, the functional response of *O. bellus* was compared using the CI value.

The step-by-step codes for implementing the “frair” package and graphical functions for the R software v4.4.2 are presented in the Appendix A.

### 2.4. Rearing Optimization

The data resulting from the functional response for each parasitoid study were used to determine the expected number of discarded hosts (*De*), which is equal to the difference between the number of hosts exposed (*N*_0_) and those expected to be parasitized (*N_e_*), calculated by the functional response at different densities. Thus, the expected proportion of discarded hosts (*P_d_*) was calculated by dividing *De* by *N*_0_. These discarded host values were calculated without considering a possible conspecific aggregation effect on parasitism. Therefore, they should be regarded as an approximation of the individual behavior of the parasitoid species addressed in this study and should be empirically validated in the future.

## 3. Results

### 3.1. Functional Response Analysis

The non-linear curve fitted for *G. pelleranoi* showed significant evidence of a Type III functional response, as the parameter *q* differed significantly from 0 (*p* = 0.001). However, according to the AIC criteria, a flexible response (“*flexpnr*”) was the best model for *G. pelleranoi*. The other three parasitoid species were significantly fitted to a Type II functional response. Nonetheless, the AIC criteria suggest that the generalized model (“*flexpnr*”) is the best for *C. haywardi* and *D. crawfordi*, matching the *p*-values close to 0.05 (*p* = 0.063 and *p* = 0.095, respectively). On the other hand, *O. bellus* was the only species that most closely fitted to “*rogersII*” (*p* = 0.727) (Table 1).

The parameters *a* (for Type II response), *b* (for *flexpnr* type response), and *h* (in both responses) involved in each type of functional response for the four parasitoid species expressed significant *p*-values, which highlighted the importance of their inclusion in the models, supporting the goodness of fit (Table 2). The handling times (*h*) recorded for *C. haywardi* and *D. crawfordi* were significantly similar (*z* = 0.759, *p* = 0.448), but significantly different from that of *G. pelleranoi* (*z* = 17.135, *p* < 0.001, and *z* = 9.823, *p* < 0.001, respectively). However, *O. bellus* had the highest handling time compared to the other three parasitoid species (Table 2).

*Coptera haywardi*, *D. crawfordi*, and *O. bellus* reached the asymptote between 15 and 40 larvae/puparia densities. In turn, *G. pelleranoi* increased the number of larvae attacked until the maximum offered host density was reached, while not reaching the asymptote (Figure 1). *Coptera haywardi* had the highest expected number of attacked hosts at density of 15 puparia over the other three parasitoid species (Figure 1). Nevertheless, *G. pelleranoi* reached the highest expected number of hosts attacked at densities ranging from 15 to 100 larvae (Figure 1). At 1–20 host densities, the confidence intervals of the functional response curve displayed by both *D. crawfordi* and *O. bellus* exhibited a high overlap. Thus, there were no significant differences between the attack rates at those host density ranges. A similar situation occurred with the overlapped confidence intervals of *D. crawfordi* and *C. haywardi* functional response curves at ranges between 60 and 100 host densities (Figure 1).

### 3.2. Rearing Optimization

At 1–5 hosts (puparia) per *C. haywardi* female, the expected percentage of discarded hosts ranged between 25 and 27%. At higher densities, e.g., 10–15 puparia per parasitoid, the percentage of discarded hosts significantly increased, ranging from 48% to 61% (Figure 2A). In *D. crawfordi* at 1–15 hosts (larvae) per female, the expected percentage of discarded hosts ranged between 82 and 85% (Figure 2B). *Opius bellus* exhibited a similar pattern to the one for *D. crawfordi*, with over 75% of hosts discarded at any offered host density (Figure 2C). *Ganaspis pelleranoi* recorded a decreasing percentage of discarded hosts between 72% and 60% at 1–20 hosts per parasitoid female (Figure 2D). The highest percentage of hosts discarded was 76% when 100 hosts were exposed to the parasitoid (Figure 2D). The complete table with the data presented in Figure 2 is in the Appendix A.

## 4. Discussion

The assessment of behavioral responses in parasitoids, mainly those related to their searching and parasitizing abilities, is a subject that must be carefully evaluated in biological control programs. This issue is important from an ecological perspective related to the host–parasitoid trophic system [50]. The above applies to both introduced parasitoid species [51] and the native resident parasitoid assemblage [42,43]. In this context, the functional response helps assess how effective a particular parasitoid species might be in controlling a target pest population [52]. Therefore, the results of the current study about the functional response analyses of the native parasitoids *C. haywardi*, *D. crawfordi*, *O. bellus*, and *G. pelleranoi* mainly provided valuable insights into their potential as biological control agents on *A. fraterculus* pest populations. In this regard, three findings can be highlighted: (1) the larval parasitoid *G. pelleranoi* displayed a flexible functional response biased to a Type III response, rather than a Type II response as shown by the other three parasitoid species; (2) *G. pelleranoi* had a handling time substantially lower than the other tested parasitoid species; and (3) the pupal parasitoid *C. haywardi* and *G. pelleranoi* had the highest attack rates, but with differences between them based on tested host densities. This study also provides a fourth finding analysis related to the expected proportion of discarded hosts by each parasitoid species, which could be useful under lab rearing conditions, suggesting some female wasp–host ratios. This last finding should be taken just as an individual behavioral approach because no mutual interference of conspecific females was measured, as in Poncio [42] and Clemente [43], and further empirical studies are needed.

The first finding showed an interesting functional response of *G. pelleranoi* differing from those exhibited by the other three tested parasitoid species. The Type III functional response is a trait of natural enemies that may learn to focus on their resource as its abundance increases [53]. The Type III or sigmoid functional response depends on host/prey density until a given threshold. Therefore, natural enemies with a Type III functional response can contribute to the system’s stability when average host/prey densities remain below the threshold [51,54]. The Type II functional response, exhibited by *C. haywardi*, *D. crawfordi*, and *O. bellus*, is the most common among parasitoid species, and it implies that the parasitism rate consistently decreases with increasing host density [55]. However, experimental conditions may affect the type of functional response, such as the arena size or the host exposure time. A Type III functional response may even occur by decreasing the time spent manipulating hosts or increasing search rates as a response to increasing host density [55]. Also, the functional responses between Type II and III may change due to different host species, host plants, and their cultivars [55,56]. Moreover, the transition from a Type II to a Type III curve can occur through a consumer learning process, but only when this phenomenon affects the attack rate [57]. Therefore, the data recorded in the current study on the types of functional response are only preliminary and relate to its particular conditions. Also, it makes new trials with the four parasitoid species possible under different factors.

The second finding, closely linked to the previous one, pointed to *G. pelleranoi* as having the best performance in interacting with the host, as this figitid parasitoid requires less time to search for, find, and parasitize a host than the other three assessed native parasitoid species. This statement is important because parasitoids whose oviposition behaviors reduce host handling time may be enhanced when the oviposition process is disturbed due to various factors [58]. Therefore, the host handling time can help check whether a parasitoid efficiently regulates the target host population [52]. Within this framework, the type of functional response and values of its parameters, such as handling time, may provide the first information about the parasitoid’s success as a biological control agent [52,55].

The third disclosure showed substantial differences among the tested parasitoid species’ ability to search for and successfully parasitize the host. Attack rates exhibited by *C. haywardi* and *G. pelleranoi* at different host densities outperformed the other two evaluated parasitoid species. Both *G. pelleranoi* and *C. haywardi* exhibited high attack rates, although for the latter species, the attack rate differed from previously reported data [43]. However, there were strong contrasts between the two parasitoid species regarding attack rate. *Coptera haywardi* needed fewer hosts than *G. pelleranoi* to achieve a better performance. *Coptera haywardi* performed well in parasitizing puparia at low host densities, while *G. pelleranoi* did so at high larval host densities. A high attack rate is a strong predictor of the efficiency of a consumer in finding and attacking its prey/host [37]. Therefore, the attack rate is a key parameter in functional response models [59]. According to Bruzzone et al. [57], the attack rate may be more important than handling time, since the type of functional response may vary only when strong learning occurs in the attack rate, not in handling time. The last point means that if learning occurs only in terms of the attack rate, the consumer becomes more efficient at finding and capturing its resource, but its handling time remains constant, so the type of functional response may change.

Both *D. crawfordi* and *O. bellus* parasitized a few *A. fraterculus* larvae at any host densities. The low attack rates exhibited by the two braconid species indicated a slower initial increase in parasitism as host larval density increased. A slower rate of prey/host consumption compared to higher attack rates may be influenced by several factors, such as handling time, learning, and/or environmental conditions [59]. Also, the searching behavior of these opinae parasitoids is different to *G. pelleranoi* and *C. haywardi*, see [11,40], which could be another point influencing the results obtained. Additionally, certain ecological factors related to reproductive population strategies may influence the results observed in these two native parasitoids. However, further studies are needed to confirm this.

The specific results found for *C. haywardi* show that the parasitoid performs better at low host densities. This condition, combined with the possibility of using *C. haywardi* in conjunction with the exotic parasitoid *D. longicaudata*, makes it interesting because it is known that these parasitoids demonstrate an additive interaction, increasing overall parasitism [60]. However, under major conditions of the shortage of non-parasitized pupae, *C. haywardi* might act as a hyperparasitoid on advanced instar *D. longicaudata* larvae [61]; thus, careful plans are needed for the joint use of these parasitoid species.

Concerning the fourth finding, the expected percentages of discarded hosts by *C. haywardi* and *G. pelleranoi* were lower than those recorded for *D. crawfordi* and *O. bellus* at any host densities exposed to parasitoids. Therefore, *C. haywardi* and *G. pelleranoi* were highly efficient at host use, because the level of successful parasitism is still higher compared to other species. Thus, the number of discarded hosts is a parameter that may improve the rearing of such parasitoids on *A. fraterculus*. This parameter would allow for adjusting the parasitoid-to-host ratio more accurately, as it helps minimize host loss. For instance, the current study showed that 1–5 *A. fraterculus* puparia and 10–20 host larvae per female parasitoid for 24 h would be enough to suit experimental rearing of *C. haywardi* and *G. pelleranoi*, respectively. The analysis of the discarded hosts does not reflect the possible conspecific interaction in rearing conditions. However, it is a preliminary approach for optimizing the parasitoids’ artificial rearing that must be corroborated in further empirical studies, showing that functional response analysis is suitable not only to select potential agents for biological control, but also to make better estimates of parasitism among such candidates [62].

Meanwhile, intraspecific interference and superparasitism are limitations for directly applying the analysis of discarded hosts under artificial rearing conditions [39,40,63]; this information, plus the attack rate, may help adjust parasitoid release rates based on the estimated densities of wild flies in a particular area [64].

Given all these findings, it is essential to point out that both parasitoids can also naturally attack and successfully develop in the exotic *C. capitata* [4]. In addition, *G. pelleranoi* can adapt to different environments where fruit trees are infested by either *C. capitata* or *A. fraterculus*. This species is widely distributed from Florida (USA), southern Mexico, and Central and South America, including southern Bolivia’s subtropical rainforest environments in northern and semiarid regions in the northwestern part of Argentina [20].

## 5. Conclusions

The present study provides critical insights into the host–parasitoid interactions involving four native parasitoid species and their hosts, specifically the larval and pupal stages of the dipteran *A. fraterculus*. This research represents the first detailed assessment of the functional responses of these species to act as potential biological control agents against this economically significant fruit pest in South America.

Our results indicate that *G. pelleranoi* and *C. haywardi* are the most promising candidates for suppressing *A. fraterculus* populations, providing valuable baseline information for the optimization of these parasitoids for laboratory rearing and the future development of large-scale programs. Notably, these two parasitoids target different developmental stages of the host—*G. pelleranoi* parasitizing larvae and *C. haywardi* parasitizing pupae—thus offering complementary control potential under both laboratory and field conditions. These findings suggest the feasibility of integrated biological control strategies targeting *A. fraterculus* in fruit-producing regions of northern Argentina, with possible applicability in other areas of Latin America.

## Figures and Tables

**Figure 1 insects-16-00919-f001:**
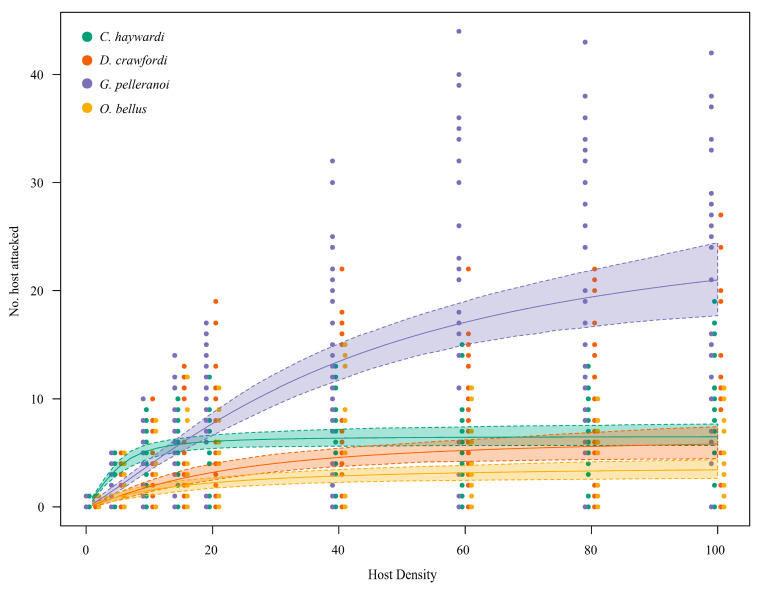
Functional responses of the four Neotropical-native parasitoid species (*Coptera haywardi*, *Doryctobracon crawfordi*, *Ganaspis pelleranoi*, and *Opius bellus*) attacking *Anastrepha fraterculus*. The dot plot shows the raw data for each parasitoid species (experimental observations), and the solid lines show the expected number of attacked hosts predicted by the models. Dashed lines are the 95% confidence intervals.

**Figure 2 insects-16-00919-f002:**
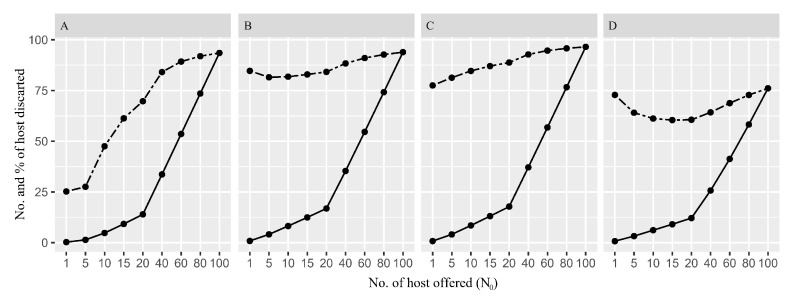
Number and percentage of the expected discarded hosts at each *Anastrepha fraterculus* puparia/larvae density offered to the Neotropical-native parasitoid species, such as (**A**) *Coptera haywardi*, (**B**) *Doryctobracon crawfordi*, (**C**) *Opius bellus*, and (**D**) *Ganaspis pelleranoi*. The continuous and dashed lines show the number and the percentage of expected discarded hosts at each host density offered to the parasitoids, respectively.

**Table 1 insects-16-00919-t001:** Recorded functional response (FR) for *Coptera haywardi*, *Doryctobracon crawfordi*, *Ganaspis pelleranoi*, and *Opius bellus*. Adjusted functional responses (FR fitted) selected by the Akaike Information Criterion (AIC), the differential Akaike Information Criterion (dAIC), and the *q* scaling exponent defining the functional response type (Type II: *q* = 0, Type III: *q* > 0, and Type I: *q* and *h* = 0), and *p*-values at α = 0.05 are shown. Models marked with (*) are the selected functional responses.

Parasitoid Species	FR	FR Fitted	AIC	dAIC	*q*	*p* < 0.05
*C. haywardi*	Type II	flexpnr	1096.814 *	0.000	0.386	0.063
		rogersII	1098.789	1.975		
*D. crawfordi*	Type II	flexpnr	3216.193 *	0.000	0.210	0.095
		rogersII	3217.234	1.041		
*G. pelleranoi*	Type III	flexpnr	3231.265 *	0.000	0.242	0.001
		hassIIInr	3237.161	5.896		
		rogersII	3239.820	8.555		
*O. bellus*	Type II	rogersII	1347.659 *	0.000	0.069	0.727
		flexpnr	1349.531	1.872		

**Table 2 insects-16-00919-t002:** Estimated parameters (*b* = constant, *q* = scaling exponent, *h* = handling time, and a = attack rate [=b*Ne^q^]) of the functional responses of the parasitoid species *Coptera haywardi*, *Doryctobracon crawfordi*, *Ganaspis pelleranoi*, and *Opius bellus*. SE: standard error, z-value from the normal distribution table, and *p*-value with α = 0.05. (*) indicate significant difference from zero.

Parasitoid Species	Parameters	Value [CI]	*SE*	*z*-Value	*p*-Value
*C. haywardi*	*b*	1.555 [0.914–3.27]	0.430	3.622	2.20 × 10^−4^ *
	*q*	0.386 [0.296–0.686]	0.208	1.860	0.063
	*h*	0.153 [0.114–0.172]	0.007	22.835	2.20 × 10^−16^ *
*D. crawfordi*	*b*	0.178 [0.048–0.391]	0.050	3.554	3.76 × 10^−4^ *
	*q*	0.210 [0.236–0.821]	0.126	1.671	0.095
	*h*	0.150 [0.060–0.949]	0.012	12.099	2.20 × 10^−16^ *
*G. pelleranoi*	*b*	0.328 [0.139–0.638]	0.064	5.113	3.17 × 10^−7^ *
	*q*	0.242 [0.067–0.586]	0.077	3.135	1.72 × 10^−3^ *
	*h*	0.036 [0.017–0.049]	0.002	13.066	2.20 × 10^−16^ *
*O. bellus*	*a*	0.274 [0.142–0.511]	0.044	6.247	4.19 × 10^−10^ *
	*h*	0.255 [0.169–0.349]	0.021	12.028	2.20 × 10^−16^ *

## Data Availability

The data set is available at CONICET Digital Data Base, URI: http://hdl.handle.net/11336/256075 (Submission date: 13 March 2025).

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
