# Peer review of "Insights into the Functional Responses of Four Neotropical-Native Parasitoids to Enhance Their Role as Biocontrol Agents Against Anastrepha fraterculus Pest Populations"

_insects, 2025, doi:10.3390/insects16090919_

Round 1
Reviewer 1 Report
Comments and Suggestions for Authors
The manuscript titled “Insights into the functional response of four Neotropical-native parasitoids for enhancing their role as biocontrol agents against Anastrepha fraterculus pest populations” presents a comparative evaluation of four parasitoid species commonly associated with Anastrepha spp., focusing on their functional response to varying host densities. The text is generally well structured, the analyses were adequately performed, and the results were adequately interpreted. English is good.
The study revealed that Ganaspis pelleranoi exhibited a Type III functional response and demonstrated the highest efficiency at higher host densities, supported by its significantly shorter handling time (h = 0.036; p < 0.001 compared to C. haywardi and D. crawfordi), indicating superior parasitization capacity. In contrast, Doryctobracon crawfordi and Opius bellus showed consistently high levels of host discard regardless of density. These findings emphasize the superior potential of G. pelleranoi and Coptera haywardi in augmentative or conservation-based biological control programs targeting A. fraterculus.
However, I have a few comments and suggestions to make the work even better
Lines 154-161: This part is not entirely clear to me: are the authors talking about studies that examined how females of the same species of native braconids interfere with each other when parasitizing (mutual interference under conditions of mass breeding. Or females of different species?
Lines 169-174: If the main focus of the study is Anastrepha fraterculus, then the formulation of the objective in this paragraph is not optimal, as it places Ceratitis capitata on equal footing with A. fraterculus, even though the rest of the abstract and text clearly indicate that A. fraterculus is the central subject. In my opinion, the final sentence is too broad and generic ("...to suppress pest fruit fly populations in Argentina, as well as in the Americas")…
Reformulation of goals: The study results stem from ongoing research, initiated in 2010, focused primarily on the colonization, mass rearing, and augmentative release of Neotropical-native parasitoids targeting Anastrepha fraterculus in fruit-growing regions of Argentina. Findings are discussed in the context of selecting effective native species to support environmentally sustainable strategies for controlling A. fraterculus populations, with potential implications for broader application in the Americas.
Lines 333-334: In the section on "discarded hosts", G. pelleranoi is shown to be effective, which is certainly in line with previous findings. However, the authors claim: "At 100 hosts, G. pelleranoi discarded 76% (fig. 2D)" - which is a high percentage for this species, and then Lines 424–426 the authors say: "...the expected percentages of discarded hosts by C. haywardi and G. pelleranoi were lower than those recorded for D. crawfordi and O. bellus at any host densities..."
Given that G. pelleranoi shows high percentages of rejected hosts at the highest densities, the authors could emphasize more clearly that these results refer to relative values, while the absolute number of successful parasitisms is still higher compared to other species. Introducing this note would contribute to a better understanding of the true efficiency of the species in the context of mass parasitism.
Lines 370-371: It is confusingly written here, the difference between type II and type III (The authors say that Type II implies that parasitism consistently decreases with increasing host density). Why is it controversial? well, that's not entirely true. In a type II functional response, the rate of parasitism per individual decreases with increasing host density (because handling time increases), but the total number of parasitized hosts increases and reaches a plateau. It is not correct to say that "parasitism decreases with increasing density" in absolute numbers. This wording should be corrected to accurately reflect a type II response.
From Figure 1, it is evident that G. pelleranoi proved to be the most efficient parasitoid at higher host densities, as it did not reach an asymptote even at the maximum host density offered, indicating a greater parasitisation capacity. C. haywardi was the most effective at lower densities (15 pupae), while D. crawfordi and O. bellus showed similar attack rates within the 1–20 host range, as well as at higher densities (60–100), where their functional response curves overlapped with that of C. haywardi. These findings highlight the varying efficiency of each species depending on host density. Also, for mass rearing optimization, G. pelleranoi demonstrated the lowest and most stable percentage of discarded hosts across a range of host densities, making it the most promising candidate for efficient large-scale production.
Lines 402-403: Imprecise statement about "Attack rate may be more important than handling time, since the type of functional response may vary only when strong learning occurs in attack rate, not in handling time learning." This is a controversial and simplistic claim. Both parameters are integral to understanding success, and their importance depends on the context of host density, interaction duration, and parasitoid species.
Lines: 419-423: In the sentence: "...C. haywardi in conjunction with the exotic parasitoid D. longicaudata... demonstrate a positive interaction, increasing overall parasitism [60]." Here the term "positive interaction" is very vague. It is not clear whether it is a matter of complementary timing of attacks, non-overlapping niches, different developmental stages or some synergy. In literature, the simultaneous use of several species of parasitoids can often cause competition, superparasitism, or interference, so it is necessary to specify which positive interaction is recorded, and whether it is really a synergy or just a collective effect.
Finally, the conclusion is clear, concise and well structured. It highlights the key findings of the study and provides recommendations that are directly related to the results obtained. It is focused on the target species (A. fraterculus), clearly identifying the two strongest candidate species (G. pelleranoi and C. haywardi), noting that they attack different developmental stages, which justifies their complementary use. It also includes the applicability of the findings, both in laboratory and field conditions, and opens up the possibility of wider regional application in Latin America.
Possibly but not necessarily, add a note about the potential for mass production and rearing optimization, as those aspects are also analyzed in the study.
Author Response
| Please see the attachment. |

Reviewer 2 Report
Comments and Suggestions for Authors
Nunez-Campero et al. Insects Review 08/05/25
The research evaluated four indigenous parasitoids, one pupal and three larval, for mass production and augmentative biological control of Anastrepha fraterculus. The parasitoids, C. haywardi, G. pelleranoi, 177 D. crawfordi, and O. bellus were reared on Anastrepha fraterculus for approximately 42 generations. For the study, a mated female of each parasitoid species was caged for 24 hours with cohorts of 1, 5, 10, 15, 20, 40, 60, 80 or 100 larvae or pupae depending on the stage that is parasitized. Each host cohort was held in an individual container and the number and sex of parasitoid offspring and of non-emerged puparia were recorded. The data were analyzed using an R package called “FRAIR” was used to compare functional response parameters (attack rate, handling time).
The manuscript is interesting and contains useful information. Apparently, the use of FRAIR to assess the functional responses of the four parasitoid species on A. fraterculus has advanced augmentation biological control of this very damaging pest. The manuscript has very few writing errors but some clarification is needed to understand how the functions were measured, e.g., handling time, in addition to the FRAIR calculations. The introduction and discussion sections seem too long and involved for such a simple laboratory study. Some of the limitations are mentioned in the discussion, including the lack of parasitoid interactions. The results that G. pelleranoi and C. haywardi are the most promising candidates for suppressing A. fraterculus populations could be presented in a considerably shorter manuscript.
Specific suggestions for improving the manuscript are as follows:
Line 50. natural enemy feeding parasitization rate at increasing host density.
Line 62. “(a) G. pelleranoi showed a flexible functional response, biased to a Type III rather than a Type II”. The results “showed significant evidence” of a Type III functional response (Line 282). It seems that the Type III functional response should be considered tentative through the manuscript.
Line 146. “Although the importance of the functional response is not debatable, the methods used to calculate the parameters that define it and allow comparison between different responses could be a cause for discussion.” Consider deleting these kinds of sentences that do not convey pertinent information.
Line 152. “which can be detailed in scientific papers, ensuring replicability. This is an example of a phrase that does not convey pertinent information. These kinds of phrases can be deleted from the manuscript.
Lines 169-174. This paragraph could be deleted without reducing the value of the manuscript.
Line 177. Where were the parasitoids collected. How were they identified?
Line 210. 5-cm diameter × 1-cm high
Line 211. “larvae were placed between screens.” More detail is needed. Could the parasitoid female and larvae interact normally?
Line 213. After exposure, host larvae and puparia were kept in individual 500 mL hinged plastic cups? Were the cohorts of exposed larvae and pupae kept in individual containers?
Line 219. The parameters measured should be listed and described here, e.g., handling time, attack rate, number of hosts discarded, etc.
Line 250. Connect sentence parts.
Line 259-263. “Models generated by each parasitoid species under study were compared. No direct statistical comparison between different parasitoid functional responses was possible.” Please clarify what these sentences mean.
Line 321. Figure 1. Add italics to scientific names.
Line 338. Figure 2. Add italics to scientific names. The x-axis should be N exposed and % hosts discarded. Or, is No should be described.
Line 361. As advised in the discussion, parasitoid female interference is an important consideration in the parasitism of hosts.
Line 366. It seems that G. pelleranoi had already learned how to efficiently parasitize A. fraterculus due to generations in rearing on this host.
Line 444. Infested by either. Insects infest, pathogens infect.
Author Response
Please see the attachment.+

Reviewer 3 Report
Comments and Suggestions for Authors
You have done excellent work and presented it a very well written manusceipt. I have a few minor questions or commens:
Line 142. Restructuring?
157-161. An awkward sentence. Testructure
177-178. Deelete the scietific names- they follow in the next sentence.
186-87. Not clear how long they were exposed.
231. This does not make sense to me (9 different host densities each exposed for 41+32+ 5+21 times?)
250. broken line
Table 3, Fig.1 and Fig 2S. Scientific name in italics
